# Epithelial Sodium Channel Alpha Subunit (αENaC) Is Associated with Inverse Salt Sensitivity of Blood Pressure

**DOI:** 10.3390/biomedicines10050981

**Published:** 2022-04-23

**Authors:** Peng Xu, Anastasia V. Sudarikova, Daria V. Ilatovskaya, John J. Gildea, Mahabuba Akhter, Robert M. Carey, Wei Yue, Pedro A. Jose, Robin A. Felder

**Affiliations:** 1Department of Pathology, University of Virginia Health System, Charlottesville, VA 22903, USA; px3x@virginia.edu (P.X.); jjg5b@virginia.edu (J.J.G.); wy9c@hscmail.mcc.virginia.edu (W.Y.); 2Division of Nephrology, Department of Medicine, Medical University of South Carolina, Charleston, SC 29425, USA; anastasia.sudarikova@gmail.com (A.V.S.); dilatovskaya@augusta.edu (D.V.I.); 3Institute of Cytology, Russian Academy of Sciences, 194064 St. Petersburg, Russia; 4Department of Physiology, Medical College of Georgia, Augusta University, Augusta, GA 30912, USA; 5Division of Endocrinology and Metabolism, Department of Medicine, University of Virginia Health System, Charlottesville, VA 22903, USA; ma5u@hscmail.mcc.virginia.edu (M.A.); rmc4c@hscmail.mcc.virginia.edu (R.M.C.); 6School of Medicine and Health Sciences, The George Washington University, Washington, DC 20052, USA; pjose@mfa.gwu.edu

**Keywords:** aldosterone, ENaC, inverse salt sensitivity, patch clamp, salt resistance, salt sensitivity

## Abstract

Salt sensitivity of blood pressure (BP) refers to an increase in BP following an increase in dietary salt, which is associated with increased incidence of cardiovascular disease and early death. However, decreased sodium intake also increases mortality and morbidity. Inverse salt sensitivity (ISS), defined as a paradoxical increase in BP on a low-salt diet, about 11% of the population, may be the cause of this phenomenon. The epithelial sodium channel (ENaC) is a major regulator of sodium reabsorption in the kidney. In this study, human renal tubular epithelial cells (hRTC) were cultured from the urine of phenotyped salt study participants. αENaC expression was significantly lower in ISS than salt resistant (SR) hRTC, while ENaC-like channel activity was dramatically increased by trypsin treatment in ISS cells analyzed by patch clamp. αENaC expression was also decreased under high-salt treatment and increased by aldosterone treatment in ISS cells. Moreover, the αENaC variant, rs4764586, was more prevalent in ISS. In summary, αENaC may be associated with ISS hypertension on low salt. These findings may contribute to understanding the mechanisms of ISS and low salt effect on morbidity and mortality.

## 1. Introduction

The amiloride-sensitive epithelial sodium channel (ENaC) is located mostly in the apical membrane of epithelial cells of the lung, distal nephron, distal colon, and ducts of exocrine glands [1]. As a non-voltage gated ion channel, ENaC constitutively allows the flow of luminal Na^+^ ions across the apical cell membrane into epithelia cells [2,3,4]. ENaC is a heterotrimeric complex comprised of three homologous subunits, α, β, and γ [1,5]. Each subunit contains two transmembrane domains, an extracellular loop, and an intracellular N- and C-termini [5,6,7]. Renal ENaC plays an important role in the regulation of whole-body Na^+^ hemostasis and the control of blood pressure (BP). It acts as a key regulator of sodium reabsorption in distal convoluted tubule (DCT), connecting tubule (CNT), and principal collecting duct (CD) [8,9]. In the kidney, the expression and activity of ENaC in the distal nephron is under the control of aldosterone and antidiuretic hormone, which are essential regulators of BP [10,11].

Abnormalities in ENaC expression or function are associated with human pathology. Genetic deletion or loss-of-function mutations in any ENaC subunit cause pseudohypoaldosteronism type 1 (PHA-1) which is characterized by Na^+^ wasting [12,13,14,15]. Global inactivation of α, β, or γENaC causes death after birth [16,17,18]. CD-specific αENaC KO mice are able to maintain sodium and potassium balance when subjected to low Na^+^ or high K^+^ feeding [19]. However, when αENaC is knocked out in both the CD and CNT, mice develop mild hyponatremia and hyperkalemia when fed a standard diet as well as a significant reduction of body weight during Na^+^ restriction [20]. The GenSalt study reported that one rare variant of αENaC, rs4764586, was associated with an increased likelihood of salt sensitivity of BP [21]. Salt sensitivity is a quantitative trait, defined as an increase of BP in response to an increase in Na^+^ load [22,23,24,25], but not necessarily to hypertensive levels. By contrast, we found inverse salt sensitive (ISS) individuals show increased BP under low salt intake and require high salt intake in order to maintain a normal BP [24,26]. As ENaC plays an important role in regulating Na^+^ balance, we hypothesized that ENaC may differentially regulate Na^+^ transport in salt resistant (SR) and ISS individuals. In order to test this hypothesis, we measured the presence of ENaC in CD13-positive renal cells collected from the urine of our dietary salt study participants who were ultimately phenotyped for their BP response to randomized transitions between week-long high (300 mEq) and low (10 mEq) Na^+^ diets [24]. Patch clamp electrophysiology was used to measure single-channel ENaC-like activity in cells from SR and ISS individuals. The role of aldosterone in regulating αENaC was also examined.

## 2. Materials and Methods

### 2.1. Cohort Stratification

Specimens were collected from volunteer participants enrolled in an Institutional Review Board-approved dietary salt study [24]. The participants (*n* = 280) were grouped into three classes, depending on the BPs obtained after 14 days of diets prepared by a dietician in the University of Virginia Medical Center containing 10 mEq Na^+^ (as NaCl) and 300 mEq Na^+^ (Appendix A). The BP measurement was described in a previous paper [24]. Briefly, the BP was measured every 15 min, 3 times at the right arm in the sitting position during each visit. There were 7 visits in total during the 2-week diet phase. The BPs on the 4th and 7th visits, at the end of the low- or high-salt diet, were used to calculate mean arterial pressure (MAP). The change in delta (Δ) MAP was defined as the MAP on the last day of the high Na^+^ diet minus the MAP on the last day of the low Na^+^ diet. The subjects were classified into SR (64% of participants) if they had an increase in MAP of <7 mmHg on 300 mEq Na^+^ intake and a decrease in MAP of <7 mmHg on 10 mEq Na^+^ intake [24]; SS (23% of participants) had an increase in MAP ≥ 7 mmHg on 300 mEq Na^+^ intake and a decrease in MAP ≥ 7 mmHg on 10 mEq Na^+^ intake [24]; and ISS (13% of participants) had a decrease in MAP ≥ 7 mmHg on 300 mEq Na^+^ intake and an increase in MAP ≥ 7 mmHg on a 10 mEq Na^+^ intake [24,26].

### 2.2. Cells from Human Urine

Human renal tubule cells (hRTCs) were collected from urine freshly voided by participants on regular diet in our clinical study as described in our previous study [27]. The cells were then briefly rinsed once by PBS++ (D8662, Sigma-Aldrich, St. Louis, MO, USA) and centrifuged at 1000× *g* for 3 min. Finally, the cells were plated in a 6-well plate and cultured in renal epithelial cell growth basal medium (REBM, cc-3190, Lonza, Walkersville, MD, USA) at 37 °C in 5% CO_2_–95% air atmosphere [27,28]. In about 2 weeks, cell colonies were visible. The cells were immortalized by human telomerase reverse transcriptase (tert) [27]. We have previously shown that renal cells voided in the urine demonstrate a natriuretic phenotype that matches their kidney’s ability to appropriately eliminate sodium, and the cells continue to express their phenotype up to 10 years after being phenotyped [29].

### 2.3. AutoMACS Selection and Flow Cytometry

Cells were grown in 15-cm dishes to 85% confluence (~10^7^) and then trypsinized, filtered using a 30-µm filter, and centrifuged at 1000× *g* for 3 min. The pelleted cells were rinsed with cold MACS buffer (Miltenyi Biotec, Boston, MA, USA), and re-centrifuged at 1000× *g* for 3 min. The supernatant was discarded, 100 µL of MACS Buffer (130-091-221, Miltenyi Biotec) was added, and the suspension was transferred to a 1.5-mL tube with 5 µL of CD13-PE (347837, BD Biosciences, San Jose, CA, USA). After mixing, the cells, protected from light, were incubated at 4 °C for 20 min, and then 1 mL of MACS Buffer was added, mixed well, and centrifuged at 1000× *g* for 3 min. The pelleted cells were then resuspended in 100 µL MACS Buffer mixed with 10 µL anti-PE magnetic beads (Miltenyi Biotec) for 20 min at 4 °C. The cells were rinsed, collected and finally resuspended in 500 µL MACS Buffer and transferred to a 5 mL tube that was kept on ice and protected from light. The cells were purified to near homogeneity using an AutoMACS (Miltenyi Biotec) for selection, using a single column. Positive and negative fractions were measured by flow cytometry (BD Accuri C6, BD Biosciences, San Jose, CA, USA).

### 2.4. Immunofluorescence Staining

The hRTCs were plated in collagen-coated 96-well glass-bottom plates. After the cells grew to 80–90% confluence, they were fixed with 4% paraformaldehyde in PBS with or without 0.2% Triton for 5 min and stained as previous described [27]. The primary antibodies in Odyssey blocking buffer were tested: L1-CAM (ab24345, 1:500, Abcam, Cambridge, MA, USA); phycoerythrin (PE)-conjugated CD13 (1:50); rabbit anti-αENAC (SPC-403, 1:200, StressMarq, Victoria, BC, Canada); rabbit anti-βENaC (SPC-404, 1:200, StressMarq Bioscinces); rabbit anti-γENaC (SPC-405, 1:200, StressMarq); and mouse anti-mineralocorticoid R (MR, MAB4369-SP, 1:200, R&D, Minneapolis, MN, USA). The appropriate secondary antibodies were used: Alexa Fluor 488 (1:500, Thermo Fisher Scientific, Waltham, MA, USA) and Alexa Fluor 647 (1:500, Thermo Fisher). Hoechst 33,342 (735969, 1:2000, Thermo Fisher) was used to stain the nuclei.

Human kidney cortical tissue was processed and immunofluorescence stained following the previously published procedure [27]. Anti-rabbit αENAC (SPC-403, 1:200, StressMarq; ASC-030, 1:200, Alomone Labs, Jerusalem, Israel), anti-rabbit βENaC (SPC-404, 1:200, StressMarq; ab28668, 1:200, Abcam), anti-rabbit γENaC (SPC-405, 1:200 StressMarq; ab3468, 1:200, Abcam), anti-mouse L1-CAM (1:500, Abcam), LTA (FL-1321, 1:500, Vector Laboratories, Burlingame, CA, USA), secondary antibodies (anti-rabbit Alexa Fluor 488 and anti-mouse Alexa Fluor 647, 1:500, Thermo Fisher) were used.

### 2.5. Confocal Microscopy

Confocal microscopy was performed using an IX81 spinning disk confocal with both mercury and xenon light sources and Semrock hard-coated filters in a Sedat configuration (Olympus, Tokyo, Japan). Images were acquired using a 20× and a 60× UPlanSApo water immersion objectives for tissue section and cell imaging. The microscope was controlled by Metamorph software (BioVision Technologies, Exton, PA, USA), which can stich single images together to obtain a view of a larger number of cells.

### 2.6. αENAC siRNA Knock-Down in Cells

The hRTCs were plated in 6-well plates and grown to 50% confluent. Human αENAC siRNA (M-006504-01-0005, Dharmacon, Cambridge, UK) was mixed with Lipofectamine RNAiMAX (13778075, Thermo Fisher, Waltham, MA, USA) for 20 min at room temperature. The mixture was then added to the cells at a final siRNA concentration of 100 nM. The cells were incubated at 37 °C in a 5% CO_2_ incubator for 24 h. Then, the medium was changed and the cells incubated for an additional 24 h. Non-targeting siRNA (Scramble, D-001210-02-50, Dharmacon) was used as control. At the end of the additional 24 h of incubation, the cells were harvested and prepared for Western blotting.

### 2.7. Western Blot

The hRTCs (90% confluent, 6-well plate) were lysed in M-PER mammalian protein extraction buffer (78501, Thermo Fisher) and sonicated. The total protein in supernatant was measured by Bradford assay (Bio-Rad Laboratories, Hercules, CA, USA), and the proteins in the samples were separated by SDS-PAGE (4–20% gradient gel). Western blotting was performed, as previously described [27]. Rabbit anti-αENAC (HPA012743, 1:500, Sigma-Aldrich; SPC-403, 1:500, StressMarq), rabbit anti-βENaC (ab28668, 1:500, Abcam), rabbit anti-γENaC (ab3468, 1:500, Abcam), mouse anti-β-actin (sc-517582, 1:1000, Santa Cruz, Dallas, TX, USA), and secondary antibodies (anti-rabbit IRDye 800 or anti-mouse IRDye 680, both at 1:15,000, LI-COR, Lincoln, NE, USA) were used. Fluorescence was imaged using a Bio-Rad Imager.

### 2.8. SNP Selection and Genotyping

Three αENaC SNPs, rs11614164, rs3741914, and rs4764586, which are associated with salt-sensitive BP, were chosen for genotyping [21] the human gDNA from the clinical cohort [24], using TaqMan SNP Genotyping Assay (Thermo Fisher). There were 280 samples; 37 were ISS subjects, 178 were SR subjects, and 65 were SS subjects.

### 2.9. Cell-Attached Configuration of Patch Clamp

Ion currents through single channels were recorded using cell-attached configuration of patch-clamp technique, based on Axopatch 200B amplifier, Analog-Digital Interface Digidata 1550B (Molecular Devices, CA, USA) and controlled by pClamp 10.7 software [30]. A Bessel filter (300 Hz) was used during the experiments. Patch pipettes were made from borosilicate glass BF-150-110-10 using P-97 puller (Sutter Instrument, Novato, CA, USA). The currents were recorded in a range of membrane potentials from −10 to −60 mV using a gap-free protocol. Extracellular pipette solution contained (mM) 140 LiCl, 5 MgCl_2_, 10 HEPES/Tris, pH 7.3; trypsin (5 μg/mL) was added in order to increase ENaC activity. In one set of experiments Li^+^ was replaced by Na^+^ in the pipette solution. Extracellular bath solution (in the chamber) contained (mM) 150 NaCl, 5 KCl, 5 glucose, 1 CaCl_2_, 2 MgCl_2_, 10 mM HEPES, pH 7.3. To quantify the level of channel activity, we used ClampFit software processing, and the following equation was used to define Po: Po = I/Ni, where Po—channel open probability, N—number of working channels in the patch, I—the mean current determined from the amplitude histograms, i—unitary current amplitude.

### 2.10. Aldosterone Treatment

hRTCs from SR and ISS subjects were plated in 96-well glass bottom plates, and then treated overnight with 1 μM aldosterone at 37 °C (1 μM aldosterone was chosen by the concentration that provided a 50% increase in total aldosterone response). The hRTCs were fixed and stained with mouse anti-mineralocorticoid receptor antibody (R&D MAB4369-SP, 1:200) and αENAC (SPC-403, 1:200, StressMarq). Montage images of each well were obtained under 20× using a Cytation 5 image reader automatically and analyzed by Gen5 software (BioTek Instrument, Winooski, VT, USA) using MR (green) as the primary mask and αENAC (red) as the secondary mask. Cell number was counted and mean value of the fluorescence density (relative fluorescence unit RFU) was used to conduct analysis. Plasma aldosterone and plasma renin activity (PRA) were measured using clinically approved diagnostic tests available through the UVA clinical laboratory.

### 2.11. Statistical Analysis

Data were expressed as means ± SE. Comparisons within and among groups (>2) were analyzed by repeated-measures or factorial ANOVA, respectively, followed by Tukey’s (comparing every mean with every other mean), Dunnet’s (comparing each mean with the other mean in that row), or Sidak’s post hoc test (comparing each mean with the control mean in that column) using GraphPad Prism 7.0 software (GraphPad Software, La Jolla, CA, USA). Student’s *t*-test was used for two-group comparisons. Chi-square test was used for SNP analysis. *p* < 0.05 was considered as significant for all analyses.

## 3. Results

### 3.1. Characterization of the Urine-Derived Cells

Many types of cells along the nephron were exfoliated into the urine. Two distinct types of cell colonies were observed. Type I cell colonies were round and small, with a fast growth rate and have clearly defined edges. The cells in the Type I colonies exhibited a dense growth pattern (i, ii of Figure 1A). The cells in the Type II colonies were elongated and less compact than the cells in Type I colonies (iii and iv of Figure 1A). Three cell lines were stained with different proximal tubule cell (PTC) markers. Samples from different salt study participants grew varying amounts of Type I and Type II colonies. In all cell lines there were many CD13- and LTA-positive (PTC makers) immunofluorescent cells [31] (Appendix A). There were some of Na^+^-Cl^−^ co-transporter (NCC)-positive cells [31] (DCT in one sample, while some Tamm–Horsfall protein (THP)-positive cells [31] (thick ascending limb cells (TAL)) were in another sample (Appendix A). There were also L1-CAM-positive cells (CNT) and CD cells marker) (Appendix A). However, majority were CD13- or LTA-positive cells possibly because of the REBM medium favors the growth of PTC. About 90% of the cells were CD13-positive after AutoMACs sorting (Figure 1B).

### 3.2. αENaC in CD13^+^ Urine-Derived Human Renal Tubule Cells (hRTCs)

αENaC, βENaC, and γENaC were all expressed on the membrane of non-permeabilized primary CD13^+^ hRTCs and distributed in a granular pattern throughout the whole cell (Figure 2 Non-permeabilized staining, Appendix A Permeabilized staining). To further confirm if ENaC was present in PTC, human kidney cortical tissue was stained with ENaC antibodies and co-stained with LTA (PTC brush-border marker) or CD13 and the lectin L1-CAM. Both αENaC and βENaC were present in LTA-positive proximal tubule (PT) apical plasma membrane (Figure 3A,B). γENaC was evenly distributed only in CD tubules but not PT (Figure 3B), indicating these three subunits of ENaC show different expression patterns in human kidney tissue. Similar staining patterns were confirmed by another set of αENaC, βENaC and γENaC antibodies (Appendix A). Furthermore, in the CD13^+^ primary cells isolated from fresh kidney cortical tissue there was also abundant αENaC staining on the apical side of the CD13-positive cells (Appendix A).

### 3.3. Comparison of αENaC Expression in Salt Resistant (SR) and Inverse Salt Sensitive (ISS) Urine-Derived hRTCs

To determine whether ENaC was present in hRTCs obtained from individuals with different salt sensitivity phenotypes, western blot analysis was conducted in urine-derived hRTCs from SR, ISS, and SS subjects (regular cell culture medium, 142 mM Na^+^). To confirm the expression of αENaC in urine-derived hRTCs, αENaC expression was knocked down by siRNA in 3 SR and 3 ISS cell lines. Western blot analysis confirmed that a single band of human αENaC with apparent molecular size between 75 and 100 kDa was present in all cell lines and reduced in the knockdown cells (Figure 4A). Figure 4B,C and Appendix A show that the expression of αENaC was significantly reduced in ISS cells compared with SR and SS cells (ISS 0.61 ± 0.1; SR 1.31 ± 0.16; SS 1.36 ± 0.25; *n* = 8/group, ISS vs. SR, ISS vs. SS *p* < 0.05, one-way ANOVA, Tukey’s post hoc test, normalized by β-actin, then by SR1). βENaC appeared as a band with expected molecular weight between 75 and100 kDa, while γENaC appeared as two bands between 75 and 100 kDa (Appendix A). βENaC expression was not different among ISS, SR, and SS (Appendix A), while γENaC expression was greater in SS than either SR or ISS cells (Appendix A).

We further examined potential changes of αENaC expression in response to low, normal and high concentrations of Na^+^. SR and ISS cells were treated overnight with 92 mM Na^+^, 142 mM Na^+^, and 192 mM Na^+^, representing low, normal, and high salt concentrations, respectively. These concentrations were selected because when we measured sodium transport in our PTC cells, we found a linear increase in Na^+^ transport through this concentration range. High-salt treatment increased αENaC expression in SR cells (192 mM/142 mM = 1.22 ± 0.15 > 1), but decreased it in ISS cells (192 mM/142 mM = 0.82 ± 0.09 < 1; SR vs. ISS *n* = 6/group, two-way ANOVA, Sidak’s post hoc test, *p* < 0.05) (Figure 4D,E), indicating that high salt concentration may decrease αENaC expression in order to reduce salt reabsorption in ISS cells.

### 3.4. Electrical Activity of ENaC in SR and ISS Urine-Derived Cells

The activity of single ENaC channel in the plasma membrane of urine-derived hRTCs was measured with the cell-attached configuration of the patch-clamp technique. The ENaC-like channels had a similar conductance in SR and ISS cells (10.5 ± 0.7 pS and E_rev_ = 34.1 ± 1.5 mV in SR cell and 10.3 ± 0.8 pS and E_rev_ = 33.6 ± 5.5 mV in ISS cells, Figure 5E). In patch-clamp studies it is common practice to induce the activity of ENaC currents, serine protease trypsin (5 μg/mL) [32,33], a known ENaC activator, was added to the pipette solution (Figure 5A). It should be noticed that the basal ENaC-like channel activity in ISS cells was too low to be recorded without trypsin in the pipette (Po = open probability) compared with that in SR cells (Figure 5B). In another set of experiments, we tested the conductivity of the channels when Na^+^ was the main cation in the pipette solution vs. Li^+^. The current-voltage dependency (IV) showed lower single channel conductance of ENaC-like channels in SR cells with Na^+^ in the pipette solution compared to Li^+^ (from 10.5 down to 8.8 pS, Figure 5C; data obtained in independent experiments); this response is typical for ENaC. Biophysical characteristics of the obtained recordings (reversal potential, conductance) were typical for ENaC-like channels which are well-established in the literature [1,34,35,36] and did not differ between SR and ISS cells (Figure 5D,E). In both cell lines, trypsin (a protease present in the urine which is known to activate ENaC) in the patch pipette evoked an increase in single ENaC-like channel activity as measured by Po [37]. However, the Po and the number of active channels in SR and ISS cells were significantly different (Figure 6). ISS cells had higher Po than SR cells in response to trypsin both immediately (0 min) and after 4 min of activation (Figure 6C), even though the number of channels recorded at 4 min after GigaOhm seal formation was lower in the ISS than SR cells (4.7 ± 1.7 vs. 1.75 ± 0.5, *p* < 0.05 in SR vs. ISS lines, *t*-test).

### 3.5. Aldosterone and αENaC

In our salt sensitivity studies [24] plasma aldosterone level was significantly higher under low-salt diet (10 mEq NaCl diet) than high-salt diet (300 mEq NaCl diet) in all of ISS, SR, and SS subjects (Figure 7A). ISS subjects tended to have the highest aldosterone level when on the low-salt diet, however, the values were not significantly different from those of SR or SS. The mineralocorticoid receptor (MR) expression was significantly increased to a similar extent by low Na^+^ treatment in both SR and ISS cells through imaging analysis (SR, 92 mM 12,194 ± 183 vs. 142 mM 10,708 ± 129; ISS, 92 mM 12,072 ± 112 vs. 142 mM 11,309 ± 277; *n* = 4/group, *p* < 0.01, two-way ANOVA, Dunnett’s post hoc test) (Figure 7B), indicating that aldosterone may increase MR expression under low-salt diet. High Na^+^ treatment decreased MR expression in ISS but not in SR cells (ISS: 142 mM 11,309 ± 277 vs. 192 mM 10,425 ± 52; *p* < 0.01, *n* = 4/group, two-way ANOVA, Dunnett’s post hoc test) (Figure 7B). Moreover, PRA was also significantly increased under a low-salt diet, and SS PRA was significantly lower than SR (Figure 8). We then determined if the urine-derived hRTCs would demonstrate the expected aldosterone-stimulated increase in ENaC when incubated under low salt (92 mM Na^+^) [38]. After an overnight treatment of the hRTCs with 1 μM aldosterone, MR and αENaC expressions were increased in ISS cells but not in SR cells through imaging analysis (MR: ISS VEH 11,818 ± 197 vs. ISS Aldosterone 12,835 ± 178, *n* = 9/group, *p* < 0.05; αENaC: ISS VEH 4561 ± 68 vs. ISS Aldosterone 4834 ± 52, *n* = 9/group, *p* < 0.01, two-way ANOVA, Sidak’s post hoc test) (Figure 7C,D, Appendix A).

### 3.6. Gene Variant rs4764586 of αENaC and Inverse Salt Sensitivity

Gene variants in αENaC have been shown to be associated with salt sensitivity of BP [21], and we verified that the specific gene variant rs4764586 is significantly associated with ISS in our salt study as well (χ^2^ = 9.73, *p* < 0.05) (Table 1). However, the other two SNPs, rs11614164 and rs3741914, did not show any difference. In our salt sensitivity clinical study, 13% of participants (N = 37/280) had a paradoxical increase in ∆MAP ≥ 7 mmHg on a low NaCl diet in comparison with a high NaCl diet, defined as ISS, versus SR (64%) and SS (23%). The prevalence of the homozygous minor allele αENaC in the ISS group was higher compared with SR and SS (ISS, 16%; SR, 5%; SS, 11%). The odds ratio of ISS to non-ISS (SR plus SS) was 1.91 indicating the odds were 1.91 times higher for those who have rs4764586 to become ISS compare with those who were homozygous major variant (Appendix A). Urine-derived hRTCs showed the lowest αENaC protein expression in the rs4764586 homozygous minor variants and the highest in the homozygous major variants (homozygous of major variants, 1.25 ± 0.24, *n* = 9; heterozygous, 1.13 ± 0.14, *n* = 12; homozygous of minor variants, 0.47 ± 0.18, *n* = 3, Appendix A).

## 4. Discussion

Renal ENaC plays an important role in the regulation of whole-body Na^+^ homeostasis. The ENaC mediated Na^+^ reabsorption occurs predominately in collecting duct because of its abundant expression in that nephron segment. Using urine-derived renal tubule cells (RTC) and human kidney tissue sections, our current studies demonstrate that α, β, and γENaC are present in urine-derived RTC, including PTC. The immunostaining was performed by two sets of antibodies, especially the StressMarq antibodies developed by Dr. Mark Knepper [39]. Both α and βENaC were present in PT, and γENaC was present only in CD. It was found that hRTC from ISS individuals were more sensitive to aldosterone induced upregulation of αENaC expression even though the basal levels of αENaC were lower than the hRTC from SR individuals. In addition, αENaC expression in hRTC in response to high-salt treatment was in opposite directions in ISS and SR, e.g., high salt reduced αENaC level in hRTC from ISS but increased αENaC level in the cells from SR. These findings suggest that differential regulation of αENaC expression and function may be one additional mechanism that contributes to inverse salt sensitivity of BP.

RNA-seq data in mouse and rat demonstrate that αENaC is more ubiquitous and plentiful than the other two subunits [39,40]. Even though the αENaC mRNA level is much lower in PT than that in DCT, CNT, or CD, its mRNA can be detected throughout the renal tubules [40,41]. Maximal ENaC activity is dependent on all three subunits being present [42,43]. αENaC alone is sufficient to induce a Na^+^ current [1]. Some vascular smooth muscle cells in renal arteries contain only β and γ ENaC, but not the α isoform. αENaC plays an important role in pressure-induced vasoconstriction [44]. Moreover, βENaC alone has been shown to control renal arterial myogenic tone both in vitro and in vivo [45,46]. In *Xenopus*, α, β, and δ ENaC can form homomeric functional channels that show amiloride sensitive currents, and channels containing two subunits also respond to shear force [47]. Therefore, our novel finding that α and β ENaC may form a functional ENaC-like channel in renal proximal tubule cells is consistent with the pleotropic role of ENaC in renal sodium homeostasis.

Differential regulation of sodium balance contributes to an individual’s “personal salt index” [27]. Inverse salt sensitivity is characterized by lower MAP while consuming the high-sodium diet and an elevated MAP while consuming the low-sodium diet. Research has shown that both high and low sodium diets result in increased cardiovascular morbidity and mortality, and is referred to a “J”-shaped relationship [48,49,50]. However, the etiology of the harmful effect of low sodium has not been determined. We found that expression of αENaC protein in the urine-derived renal cells was significantly lower in ISS than in SR. Sodium treatment increased αENaC in SR but decreased it in ISS under 192 mM sodium. These findings indicate that the ISS with lower BP on high-salt diet may result from reduced sodium reabsorption because of less αENaC expression. Interestingly, reduction in αENaC was also observed in rats on Na^+^-replete diet compared with the rats on low-Na^+^ diet [51]. Single-channel patch clamp analysis showed that ISS cells have higher open probability of ENaC-like currents post-trypsin treatment compared with SR cells. These changes indicate that ISS cells not only have reduced levels of total αENaC but also have more silent ENaC-like channels in the membrane.

Aldosterone is dramatically increased under low salt conditions [52]. It also increases ENaC abundance and stimulates ENaC activity at the apical cell membrane through mineralocorticoid receptor (MR) and an ENaC-regulatory-complex pathway with involvement of SGK1 (serum and glucocorticoid-induced kinase 1) and GILZ (glucocorticoid-induced leucine zipper protein) [53]. In our clinical study, as anticipated, plasma aldosterone increased under low sodium diet in all participants. There was a trend that ISS individuals had higher plasma aldosterone concentrations. The in vitro study showed that aldosterone treatment upregulated MR and αENaC in PTC from ISS individuals but not from SR individuals. This could explain how low-sodium-diet-induced elevation of aldosterone causes a more dramatic increase in both MR and ENaC expression in ISS cells which leads to more efficient sodium reabsorption and increased BP. By contrast, under high salt conditions ISS individuals are protected from increased BP by lower ENaC expression and greater natriuresis. The fact that aldosterone was not significantly different when comparing ISS to SR under high-salt diet indicate that these particular experiments may have been underpowered, since ISS occurs at a lower frequency (13%) compared with SS (23%). Follow up studies will be necessary to answer this particular question. The elevated circulating aldosterone in ISS while consuming a low-salt diet coupled with the higher sensitivity to aldosterone in ISS ex-vivo urine-derived renal cells may be a diagnostic tool to identify ISS individuals. We also found an elevated PRA levels while on the low-salt diet, and SS individuals showed significantly lower PRA than SR individuals, indicating that low-salt diet is protective for SS individuals.

αENaC variants were reported to be associated with salt sensitivity of BP in a Chinese GenSalt study [21]. Those who carried variant rs4764586 had 1.36-fold increased odds ratio for salt sensitivity [21]. In our study where we defined ΔMAP ≤ −7 mmHg as ISS, rs4764586 was also associated with salt sensitivity. However, this rare variant shows increased odd ratio (1.91) in ISS, indicating that this rare variant may be associated with inverse salt sensitivity of BP as well.

Others have also demonstrated the utility of urine-derived cells. For example, the first successful culture of exfoliated urinary cells was from newborn infants reported by Sutherland and Bain [54], and subsequently multiple groups have reproduced the culture from both infant and adult urine [55,56,57,58,59]. These urine-derived cells were characterized as a heterogeneous population of different cell types, including urothelial, PT, DCT, and fibroblast-like cells [59,60]. More recently, a subpopulation of urine-derived cells was also identified as a good source of stem cells for cell therapy and tissue engineering [61,62,63,64]. All of the previous studies were performed on the primary cell culture. In our culture technique, the cells are immortalized by tert and can be sub-cultured as a stable cell line for many passages. Consistent with the previous reports [56,57,58,59], there were two cell types in most cultures and the type I cell colony with smooth edge usually grows rapidly and is easy to maintain as a cell line. In our study, we also found that there were different types of renal tubule cells, such as DCT, CD, etc. Urine-derived renal tubule cells can serve as representative models to study human renal cell physiology.

A limitation of our studies is that we did not determine the effect of high and low-salt diets on ENaC expression and activity in renal tissue. Renal tissue cannot be obtained from normal individuals as IRB approval cannot be obtained for experimental renal biopsies. Discard kidneys from transplant services have proved not to exhibit normal physiological responses related to Na^+^ homeostasis and also exhibit necrotic features. Another limitation is the sample size of urine cell lines, which is dependent on participant recruitment and urine cell culture. Furthermore, characterization of these urine-derived renal tubule cells should be performed in later passages because the cells have stem-cell characteristics and may de-differentiate after being subcultured many times [65]. Physiologic studies in rodents demonstrated that infusion of a relatively large amount of NaCl decreases the expression of ENaC [50]. Future studies of hRTCs from ISS individuals and SR controls obtained from fresh urine might yield some insight into to whether elevated Na^+^ increases or decreases ENaC expression. Further, the differential regulation of ENaC between ISS and SR also need to be further addressed. For example, Mammalian Ste20 kinase 3 (MST3) may be involved in regulating ENaC expression in these hRTCs, which maintain BP through inhibiting ENaC expression in mice [66,67].

## Figures and Tables

**Figure 1 biomedicines-10-00981-f001:**
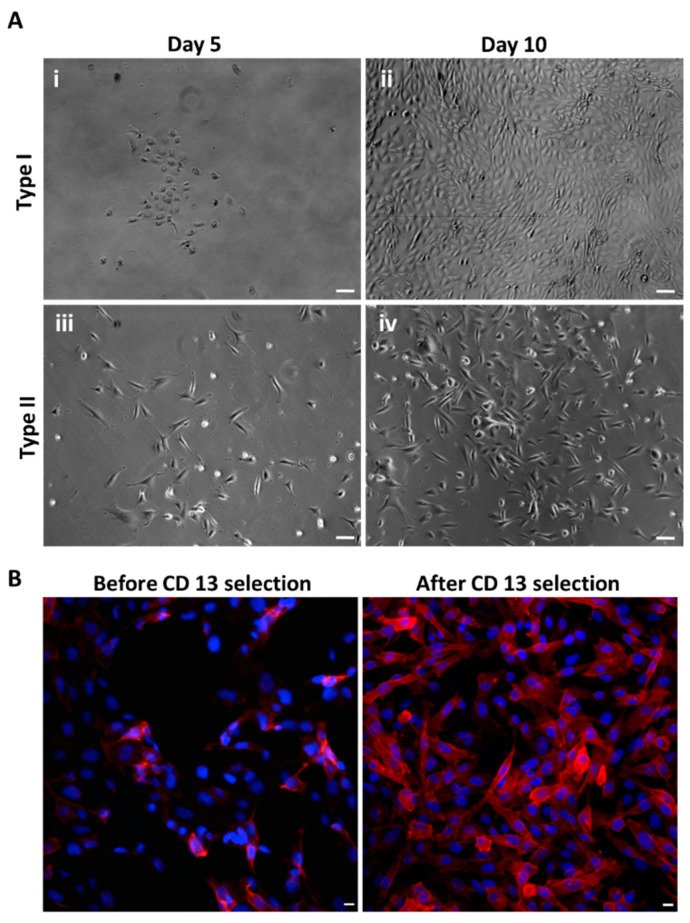
**Urine cell culture.** (**A**) Representative images of cell colonies from urinary cultures. There are two distinct types of cell colonies. (i,ii): Type I cell colony is commonly found in all the cultures. The round and relatively small cells grow fast with a smooth edge. Within the colony, the cells are compact. (iii,iv): Type II cell colony is less common than type I, but as often found in urine cell culture. The cells were long with slower growth characteristics than type I cells. They formed a loose colony. (Scale bar = 50 μm.) (**B**). Representative images of cells sorted by CD13 in red, a PTC marker. Nuclei were stained blue. (Scale bar = 10 μm.)

**Figure 2 biomedicines-10-00981-f002:**
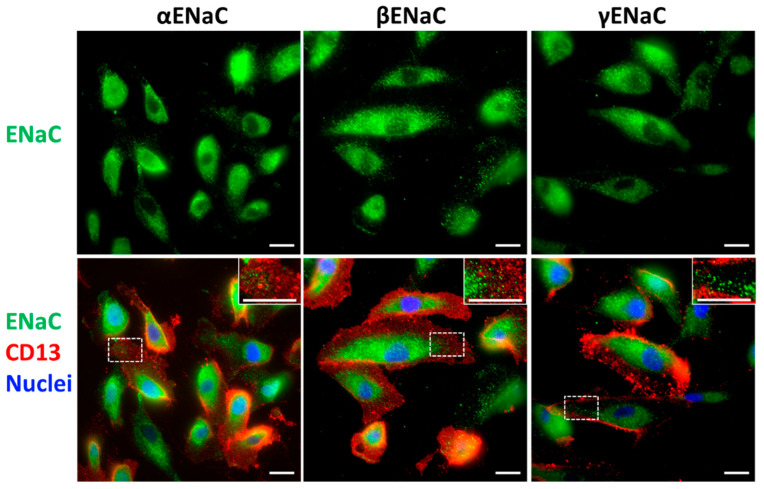
**αENaC, βENaC, and γENaC were stained in non-permeabilized primary urine-derived hRTC.** αENaC, βENaC, and γENaC were stained green. CD13 staining (red fluorescence) was seen on cell membrane as a punctate pattern. Nuclei were in blue. The detailed view was highlighted by dashed rectangle. (Scale bar = 10 µm).

**Figure 3 biomedicines-10-00981-f003:**
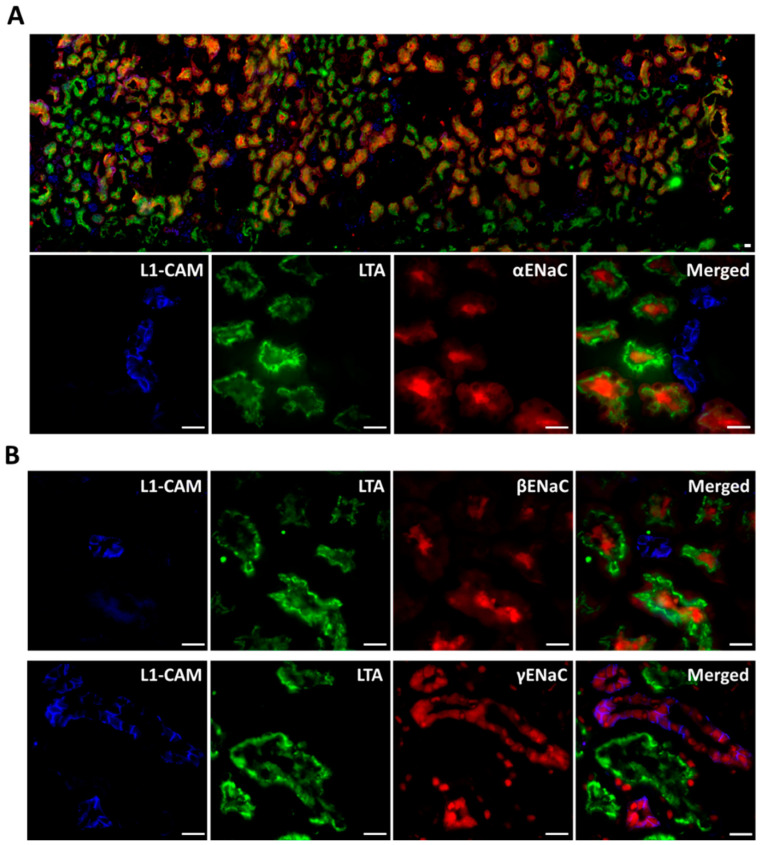
**αENaC, βENaC, and γENaC staining in human renal cortex.** (**A**) The upper panel was a montage showing the overall αENaC, L1-CAM, and LTA staining in the whole slice of the tissue; the lower panel showed the close-up images. (**B**) Representative image of βENaC and γENaC staining in kidney tissue. α, β, and γENaC were stained in red. LTA was a PTC marker in green. L1-CAM, a CD marker, was in blue. (Scale bar = 10 µm).

**Figure 4 biomedicines-10-00981-f004:**
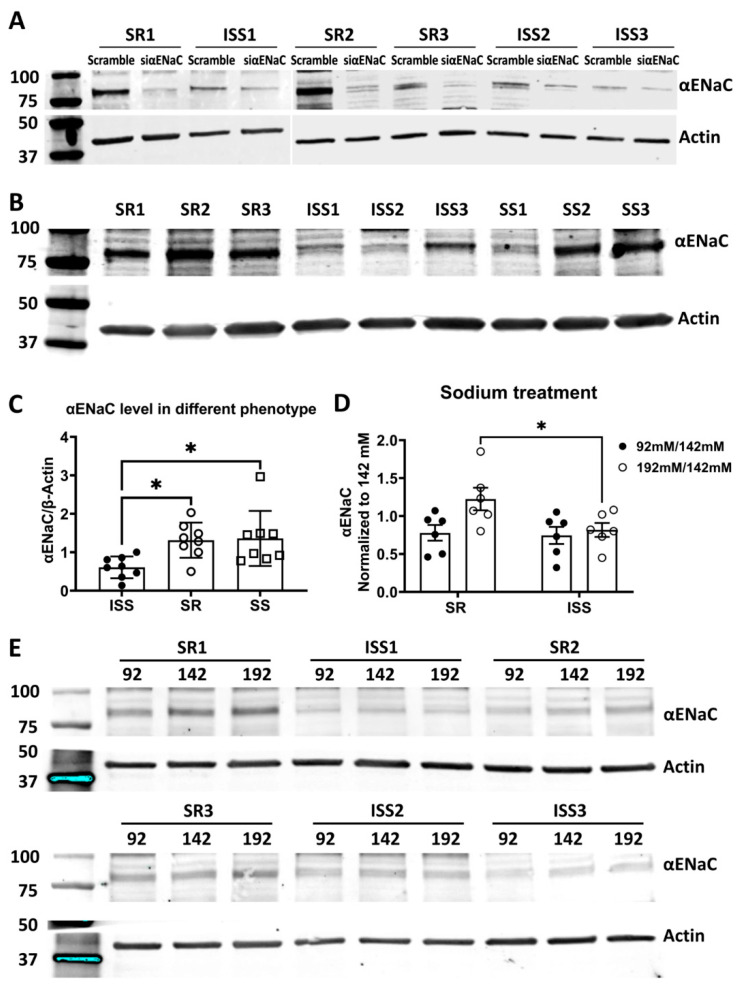
**αENaC in urine-derived renal tubule cells by Western blot.** (**A**) αENaC siRNA was used to knock down αENaC to confirm that αENaC is about 85 kDa. Scrambled RNA was used as control. (**B**,**C**) Representative αENaC Western blot in urine-derived hRTCs (0.61 ± 0.1; SR 1.31 ± 0.16; SS 1.36 ± 0.25; *n* = 8/group, ISS vs. SR, ISS vs. SS, * *p* < 0.05, one-way ANOVA, Tukey’s post hoc test, normalized by β-actin, then by SR1). (**D**,**E**) Sodium treatment in SR and ISS CD13+ renal tubule cells. The ratio of high sodium over normal sodium is significantly lower in ISS than SR (192 mM/142 mM: SR, 1.22 ± 0.15 vs. ISS, 0.82 ± 0.09, two-way ANOVA, Sidak’s post hoc, * *p* < 0.05; 92 mM/142 mM: SR, 0.78 ± 0.10; ISS 0.74 ± 0.11, *n* = 6/group). β-actin was used as loading control in all western blot analysis.

**Figure 5 biomedicines-10-00981-f005:**
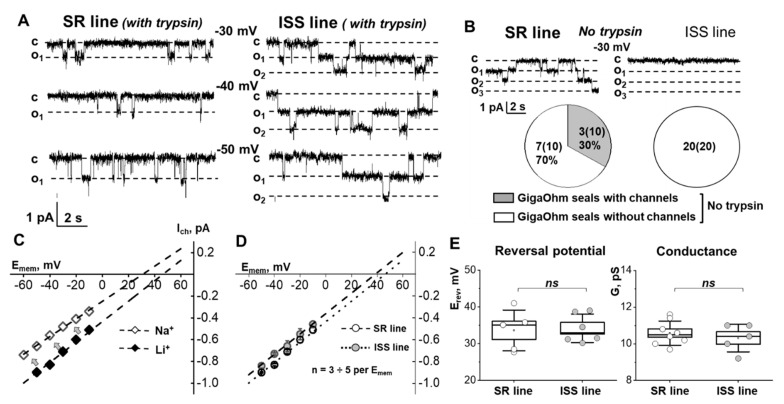
**ENaC-like channels recorded in SR and ISS cell lines**. (**A**) Representative current traces showing single ENaC-like channels’ activity in SR and ISS cell lines recoded at different membrane potentials using patch-clamp electrophysiology. (**B**) Channel prevalence in successful GigaOhm seals in SR and ISS cell lines without stimulation with trypsin. Representative current traces are shown at −30 mV. (**C**) IV curve demonstrating lower single conductance of ENaC-like channels in a SR cell line with Na^+^ as main cation in the pipette vs. Li^+^. (**D**) Cumulative IV curves for single ENaC-like channels recorded in SR and ISS cell lines. (**E**) Summary graphs demonstrating reversal potential and single channel conductance (SR line: E_rev_ = 34.1 ± 1.5 mV, *n* = 4, and 10.5 ± 0.7 pS *n* = 8 vs. ISS line: E_rev_ = 33.6 ± 5.5 mV, *n* = 6, and 10.3 ± 0.8 pS, *n* = 5, *t*-test).

**Figure 6 biomedicines-10-00981-f006:**
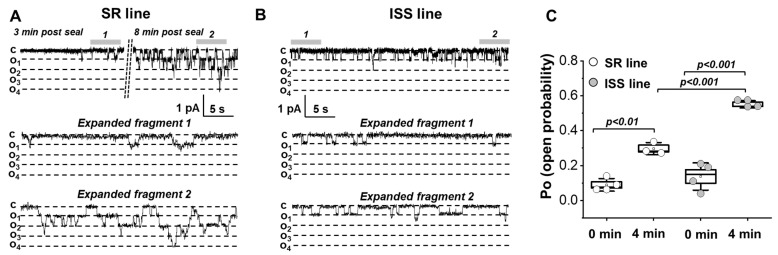
**Trypsin-triggered activation of ENaC-like channels in the SR and ISS cells**. Representative current traces showing single ENaC-like channel activity in SR (**A**) and ISS (**B**) cell recoded 1 min after GigaOhm seal formation, and 3 min and 8 min post-seal formation; pipette solution contained 2 μg/mL trypsin. Shown are representative recordings as well as expanded regions at −20 mV and −30 mV for SR and ISS lines, respectively. Scale bar is noted on the graph, c and oi denote closed and open states (inward currents are downward). (**C**) A summary graph of the open probability of the channels calculated for at least 20 s recordings immediately after the start of the activation of channels (0 min), and 4 min post-activation start (4 min) (SR: 0 min, 0.089 ± 0.018 *n* = 4 vs. 4 min, 0.297 ± 0.02 *n* = 3, *p* < 0.01; ISS: 0 min, 0.138 ± 0.039 *n* = 4 vs. 4 min, 0.551 ± 0.012 *n* = 3, *p* < 0.001, two-way ANOVA, Tukey’s multiple comparisons test).

**Figure 7 biomedicines-10-00981-f007:**
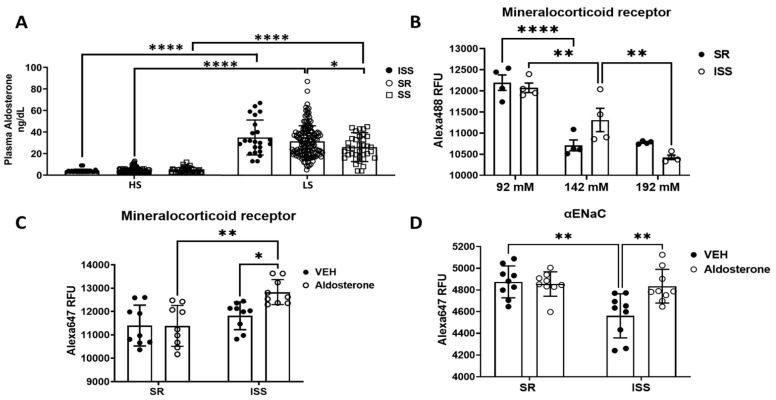
**Aldosterone level from our dietary salt study and aldosterone treatment on urine-derived renal tubule cells.** (**A**) Plasma aldosterone level of ISS, SR and SS participants on low- and high-salt diets (ISS HS, 3.8 ± 0.38, *n* = 26; ISS LS, 35 ± 3.38, *n* = 22; SR HS, 4.34 ± 0.18, *n* = 180; SR LS, 32.62 ± 1.6, *n* = 152; SS HS, 4.65 ± 0.35, *n* = 43; SS LS, 26.08 ± 2.18, *n* = 38; HS vs. LS, *p* < 0.001, two-way ANOVA, Tukey’s multiple comparisons test). (**B**) MR level under different sodium treatments in both SR and ISS cells (SR, 92 mM 12,194 ± 183 vs. 142 mM 10,708 ± 129; ISS, 92 mM 12,072 ± 112 vs. 142 mM 11,309 ± 277; *n* = 4/group, *p* < 0.01, two-way ANOVA, Dunnett’s multiple comparisons test). (**C**,**D**) Mineralocorticoid receptor (MR) and αENaC level under aldosterone treatment (1 μM) on both SR and ISS cells (MR: ISS VEH 11,818 ± 197 vs. ISS Aldosterone 12,835 ± 178, *n* = 9/group, *p* < 0.05; αENaC: ISS VEH 4561 ± 68 vs. ISS Aldosterone 4834 ± 52, *n* = 9/group, *p* < 0.01, two-way ANOVA, Tukey’s multiple comparisons test). * *p* < 0.05, ** *p* < 0.01, **** *p* < 0.001.

**Figure 8 biomedicines-10-00981-f008:**
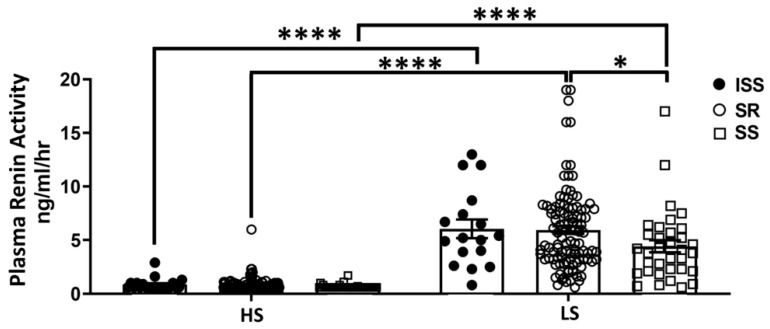
**Plasma renin activity level of ISS, SR and SS participants on low and high-salt diets** (ISS HS, 0.86 ± 0.12, *n* = 21; ISS LS, 6.05 ± 0.87, *n* = 17; SR HS, 0.8 ± 0.55, *n* = 144; SR LS, 5.94 ± 0.36, *n* = 108; SS HS, 0.66 ± 0.03, *n* = 44; SS LS, 4.43 ± 0.57, *n* = 34; HS vs. LS, *p* < 0.001, two-way ANOVA, Tukey’s multiple comparisons test). * *p* < 0.05, **** *p* < 0.001.

**Table 1 biomedicines-10-00981-t001:** Distribution of αENaC rare variant, rs4764586, in different salt sensitivities.

Salt Sensitivity	Homozygous of Major Variants	Heterozygous	Homozygous of Minor Variants	Total (*n* = 280)
**ISS**	14 (38%)	17 (46%)	6 (16%)	37 (13%)
**SR**	93 (52%)	76 (43%)	9 (5%)	178 (64%)
**SS**	38 (58%)	20 (31%)	7 (11%)	65 (23%)

χ^2^ = 9.73 *p* < 0.05.

## Data Availability

Not applicable.

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
