# Peer review of "Epithelial Sodium Channel Alpha Subunit (αENaC) Is Associated with Inverse Salt Sensitivity of Blood Pressure"

_biomedicines, 2022, doi:10.3390/biomedicines10050981_

Round 1

Reviewer 1 Report

General comments:

The authors in this manuscript investigate whether ENaC differentially regulate Na+ transport in SR and ISS individuals. They cultured human renal tubular epithelial cells (hRTC) obtained from the urine of phenotyped salt study participants and examined ENaC expression and activity, aldosterone and mineralocorticoid receptor (MR) levels, and ENaC and MR response to aldosterone treatment. The authors found that αENaC expression was significantly lower in ISS than SR hRTC, while there was a marked increase in ENaC-like channel activity after trypsin treatment in ISS cells. αENaC expression was also decreased under high salt treatment and increased by aldosterone treatment in ISS cells. In addition, αENaC polymorphism, rs4764586, was more prevalent in ISS. The results are interesting, but some key data and the authors’ interpretation/conclusion appear to be a bit problematic. My specific comments are below.

Specific comments:

Figure 2 and 3. Immunofluorescence images of ENaC subunits show that they are mainly in cytosol with little or no expression in cell membranes. It is true that the images demonstrate ENaC expression in the PT-like hRTCs, but there is basically negligible colocalization of ENaC and CD13 or LTA. This makes it difficult to validate the quantitation of ENaC expression in Figures 4 and 7.

Figure 4. Uncropped immunoblot images in the supplemental figures show too many backgrounds. While I am aware of the problem of current ENaC antibodies, the immunoblotting could have been improved with better dilutions of antibodies. siRNA knockdown showed substantial decrease in αENaC, but other bands were also decreased as shown in supplemental Figure 4A.

Figure 4E is one of the key data that test the hypothesis the authors made in the manuscript, but the number of samples is small (n=3) and the band intensities between different sodium treatments are very minor. Comparison of these immunoreactive band intensities is thus highly subjective. The authors should increase the number of samples to validate the changes. Also, the statement ‘the ratio of high sodium over normal sodium is significantly lower in ISS than SR’ is misleading because ENaC expression levels are already low in ISS.

Figure 7. Please include MR fluorescence images that were used for quantitation in Figure 7B and C, and ENaC images for quantitation in Figure 7D. The images can be submitted as supplemental materials. In addition, describe, in Methods, how fluorescence was quantitated.

Discussion. The hRTCs are largely proximal tubule cells. The data from this study show ENaC and MR expression in these cells, thus leading to the likelihood that hRTCs are not ordinary epithelial cells in the kidney tubules but instead could be cells that are aberrantly differentiated. It is true that ENaC mRNA can be found throughout the renal tubules, but this does not imply that ENaC in the proximal portion of the tubules (eg, PT) makes significant contribution to salt reabsorption. In this sense, the author’s interpretation of the data, and conclusion on ENaC association with salt sensitivity, are not strongly validated.   

Minor comments:

Supplemental Figure 7 should be presented as figure in the main manuscript.

Line 313, plasma.

Line 314, Supplemental Fig. 7.

Author Response

We are grateful that the Reviewers have found value in our work and provided very helpful recommendations for the improvement of our manuscript. Below we have carefully addressed each of the Reviewers’ critiques. In addition, we have corrected typos and errors which can be traced for you and reviewers’ review.

Reviewer 1

The authors in this manuscript investigate whether ENaC differentially regulate Na+ transport in SR and ISS individuals. They cultured human renal tubular epithelial cells (hRTC) obtained from the urine of phenotyped salt study participants and examined ENaC expression and activity, aldosterone and mineralocorticoid receptor (MR) levels, and ENaC and MR response to aldosterone treatment. The authors found that αENaC expression was significantly lower in ISS than SR hRTC, while there was a marked increase in ENaC-like channel activity after trypsin treatment in ISS cells. αENaC expression was also decreased under high salt treatment and increased by aldosterone treatment in ISS cells. In addition, αENaC polymorphism, rs4764586, was more prevalent in ISS. The results are interesting, but some key data and the authors’ interpretation/conclusion appear to be a bit problematic. My specific comments are below.

We thank the reviewer for the careful review of the manuscript.

Specific comments:

  1. Figure 2 and 3. Immunofluorescence images of ENaC subunits show that they are mainly in cytosol with little or no expression in cell membranes. It is true that the images demonstrate ENaC expression in the PT-like hRTCs, but there is basically negligible colocalization of ENaC and CD13 or LTA. This makes it difficult to validate the quantitation of ENaC expression in Figures 4 and 7.

The CD13 and LTA antibodies are mainly used as proximal tubule cell markers in Figure 2 and 3. Our intent was not to study the colocalization here but rather to show co-expreession in the tubule segment.  We have little control over the time from tissue harvesting to and delivery to our lab, and often the tubules are collapsed and microvilli shed into the lumen during transport.  We are not sure why the lectin binding epitope remains closer to the cell than the single transmembrane spanning CD13, but both can be considered apically oriented. CD13 and ENaC do not appear to be colocalized in our samples but are co-expressed in the same tubule segment without exception. In Figure 2, the cultured cells are non-permeablized and not cultured to confluence and differentiated, the ENaC staining is on the membrane with nice punctate pattern compare to the permeablized staining with heavy staining in the nuclear area in Supplemental Figure 2.  CD13 protein does appear to be able to move throughout plasma membrane and show more prominently in membrane ruffles toward the cell periphery while the ENAC subunits does appear to reside in more punctate structures resembling rudimentary microvilli, yet both protein are residing within the same cell surface plasma membrane.

  1. Figure 4. Uncropped immunoblot images in the supplemental figures show too many backgrounds. While I am aware of the problem of current ENaC antibodies, the immunoblotting could have been improved with better dilutions of antibodies. siRNA knockdown showed substantial decrease in αENaC, but other bands were also decreased as shown in supplemental Figure 4A.

Yes, the HPA αENaC antibody showed a lot of background. As you may be aware, glycosylated proteins with substantial extracellular structure migrate on SDS Page gels in more complicated sizing patterns and we believe you are right, that some of these are non-specific. But it was the only antibody working at the beginning of this study. The Alomone antibody works well on mouse tissue only (the blot was not included in the supporting document, but showed to the editor). We initially used the mouse tissue as a positive control to determine the αENaC band. In all six pairs of samples, αENaC siRNA specifically reduced the levels of this protein. Whereas, changes in irrelevant bands were not consistent. For example, reduction of non-specific bands was only seen in the SR2 knockdown sample and the bands between 50 and 75 kDa went up in SR3 knockdown sample.  Nevertheless, these unexplainable changes in irrelevant bands do not prevent us to conclude that the siRNA used in this experiment was specific for αENaC. This also suggests that a better αENaC antibody is needed.  Recently, we received and tested an αENaC antibody from StressMarq, which did produce a cleaner western blot than the HPA antibody, and is shown in Supplemental Figure 5.

  1. Figure 4E is one of the key data that test the hypothesis the authors made in the manuscript, but the number of samples is small (n=3) and the band intensities between different sodium treatments are very minor. Comparison of these immunoreactive band intensities is thus highly subjective. The authors should increase the number of samples to validate the changes. Also, the statement ‘the ratio of high sodium over normal sodium is significantly lower in ISS than SR’ is misleading because ENaC expression levels are already low in ISS.

Cell lines used in Figure 4E are well established years ago, they are quite stable. Recently we have expanded new cell lines in each salt phenotype. It will take some time to stabilize these new cell lines. I have added more samples to this experiment. The new western blot was included in supporting document Figure 4D new.

This experiment was to answer the question whether low or high salt concentration would change the levels of αENaC expression in PTC cells, and in what direction. Therefore, the levels of αENaC at normal salt served as a reference or standard. The ratio of high salt/normal salt made the salt-induced changes in αENaC comparable between the samples from different phenotypes (SR vs ISS). Our results have shown that the ratio is 0.82±0.09, less than 1 in ISS and 1.22±0.15 larger than 1 in SR. The results indicate that high salt concentration further reduced αENaC expression in ISS whereas its expression in response to high salt in SR was in opposite direction.

  1. Figure 7. Please include MR fluorescence images that were used for quantitation in Figure 7B and C, and ENaC images for quantitation in Figure 7D. The images can be submitted as supplemental materials. In addition, describe, in Methods, how fluorescence was quantitated.

The images of MR and αENaC staining was included in the supplemental Figure 7, and the detailed description was added to the method as well.

  1. The hRTCs are largely proximal tubule cells. The data from this study show ENaC and MR expression in these cells, thus leading to the likelihood that hRTCs are not ordinary epithelial cells in the kidney tubules but instead could be cells that are aberrantly differentiated. It is true that ENaC mRNA can be found throughout the renal tubules, but this does not imply that ENaC in the proximal portion of the tubules (eg, PT) makes significant contribution to salt reabsorption. In this sense, the author’s interpretation of the data, and conclusion on ENaC association with salt sensitivity, are not strongly validated.   

The main focus of this paper is that we can culture urine cells from the participants in our clinical study, and analyze these cells which are directly from different salt phenotypes. The shortcoming is that the cells may change their characteristics during culture. And we are working hard to keep the cell lines stable. Identifying ENaC in the proximal tubule cells is a new finding of this study. We understand it is unexpected. The presence of ENaC in PTC has been carefully validated by immunofluorescence staining using two different antibodies. We agree that the presence of ENaC in PT may not mean that it makes significant contribution to salt reabsorption but this would only be testable in human clinical studies or specific mouse genetic studies. Additional studies are clearly needed to determine the physiological significance of this finding but the absence of renal physiologic data in humans does not rule out this possibility. By comparison of hRTCs ENaC function and its expression in response to high salt concentration, our data provided evidences that ENaC may play a significant role in ISS. It is yet to be determined whether sodium wasting as seen in pseudohypoaldosteronism type 1, is a significant and necessary component of the newly discovered and understudied ISS phenotype in humans.

Minor comments:

Supplemental Figure 7 should be presented as figure in the main manuscript.

Line 313, plasma.

Line 314, Supplemental Fig. 7.

Supplemental Figure 7 was changed to Figure 8, and figure numbers was also changed accordingly in the context.

Reviewer 2 Report

The manuscript concluded that αENaC might be associated with ISS hypertension on a low salt diet based on the following results:

  1. ENaC expression was significantly lower in ISS than SR hRTC.
  2. ENaC like channel activity was dramatically increased by trypsin treatment in ISS cells analyzed by patch-clamp.
  3. αENaC expression was also decreased under high salt treatment and increased by aldosterone treatment in ISS cells.

The topic is of interest. The paper was well-organized. Moreover, these findings might lead to an insightful interpretation of the mechanisms of ISS on a low salt. However, I would recommend the following comments that should be addressed.

  1. Introduction: most of the results in the paper were observed in the proximal tubules. It would be helpful for the readers to follow if adding some background information about ENaC in the proximal tubules.  
  2. αENaC expression was also decreased under high salt treatment and increased by aldosterone treatment in ISS cells. I wondered if the author investigated the effect of high salt on the SS cells? Please elaborate on that.
  3. Methods: please briefly describe isolating the renal tubular cells from human urine and cite the previous publication.
  4. Please spell it out when the abbreviation first showed up. -For example, line 21, SR; line 185, PT and PTC, etc.
  5. Add the legends to Figure 3A&B. 
  6. Please be consistent. i.e., either Figure or Fig.
  7. Please carefully check and correct the Reference-for example,
    • Reference 21 should be 22.
    • Reference 24 seemed not to match the context.
    • Cite the references 63-66 in the context of the manuscript.

     8. Extra space between “had” and “an” ln line 71.

Author Response

We are grateful that the Reviewers have found value in our work and provided very helpful recommendations for the improvement of our manuscript. Below we have carefully addressed each of the Reviewers’ critiques. In addition, we have corrected typos and errors which can be traced for you and reviewers’ review.

Reviewer 2

The manuscript concluded that αENaC might be associated with ISS hypertension on a low salt diet based on the following results:

  1. ENaC expression was significantly lower in ISS than SR hRTC.
  2. ENaC like channel activity was dramatically increased by trypsin treatment in ISS cells analyzed by patch-clamp.
  3. αENaC expression was also decreased under high salt treatment and increased by aldosterone treatment in ISS cells.

The topic is of interest. The paper was well-organized. Moreover, these findings might lead to an insightful interpretation of the mechanisms of ISS on a low salt. However, I would recommend the following comments that should be addressed.

We thank the reviewer for appreciating the value of the manuscript.

  1. Introduction: most of the results in the paper were observed in the proximal tubules. It would be helpful for the readers to follow if adding some background information about ENaC in the proximal tubules.  

ENaC was mostly studied in distal tubule cells and collect duct cells. Low level of ENaC mRNA was found in an RNAseq study which was included in the discussion. This is the first report on ENaC expression and function presented in human proximal tubule cells.

  1. αENaC expression was also decreased under high salt treatment and increased by aldosterone treatment in ISS cells. I wondered if the author investigated the effect of high salt on the SS cells? Please elaborate on that.

We have not yet completed our systematic study on the human SS cells.  Distal tubule segment specific cells types are found in our urine cultures, but they are less abundant and more difficult to grow. ENAC and SS has been studied in a human genetic study but we believe it is a much more complicated cell physiologic study for us because of the perceived necessity study both connecting tubule and collecting duct cells because of the clear rodent data implicating specific distal nephron segments in ENAC knockdown studies. We strongly agree that it is a good idea to understand the effect of salt and aldosterone on the whole salt sensitivity spectra as well as the effect on both proximal and distal nephron segments in humans.

  1. Methods: please briefly describe isolating the renal tubular cells from human urine and cite the previous publication.

More details were added to the method.

  1. Please spell it out when the abbreviation first showed up. -For example, line 21, SR; line 185, PT and PTC, etc.

Abbreviations have been corrected.

  1. Add the legends to Figure 3A&B. 

More details were added to Figure 3A and B.

  1. Please be consistent. i.e., either Figure or Fig.

The error has been corrected. Fig. was used in the main context and Figure was used in the figure legends now.

  1. Please carefully check and correct the Reference-for example,
  • Reference 21 should be 22.
  • Reference 24 seemed not to match the context.
  • Cite the references 63-66 in the context of the manuscript.

The error has been corrected in the references.

  1. Extra space between “had” and “an” ln line 71.

The error has been corrected.

Reviewer 3 Report

The experiments are very interesting and the study provides valuable data. One minor comment: in the methods, the authors describe the cohort stratification based on the BP response to a high-salt diet. The authors should provide additional details. For example, how was the BP measured in this cohort? The authors could add a supplement table with the clinical characteristics of the patients to describe better the cohort stratification.

Author Response

Reviewer 3

The experiments are very interesting and the study provides valuable data. One minor comment: in the methods, the authors describe the cohort stratification based on the BP response to a high-salt diet. The authors should provide additional details. For example, how was the BP measured in this cohort? The authors could add a supplement table with the clinical characteristics of the patients to describe better the cohort stratification.

We thank the reviewer for appreciating the value of the manuscript.

The characteristics chart was added to the supplemental document as supplemental table 1. And more detailed description of blood pressure measurement was added to the method.

Round 2

Reviewer 2 Report

Extra figure in page 9? Please carefully check it.

Reviewer 3 Report

No further comments.
